

# The use of artificial substrate units to improve inventories of cryptic crustacean species on Caribbean coral reefs

Luz Verónica Monroy-Velázquez[1,*], Rosa E. Rodríguez-Martínez[1,*], Paul Blanchon[1] and Fernando Alvarez[2]

[1] Instituto de Ciencias del Mar y Limnología, Universidad Nacional Autónoma de México, Puerto Morelos, Quintana Roo, México

[2] Colección Nacional de Crustáceos, Instituto de Biología, Universidad Nacional Autónoma de México, Mexico City, Mexico City, México

* These authors contributed equally to this work.

Corresponding authors
Luz Verónica Monroy-Velázquez,
plesionika.vmv@gmail.com
Rosa E. Rodríguez-Martínez,
rosaer@cmarl.unam.mx

## ABSTRACT

Motile cryptofauna inhabiting coral reefs are complex assemblages that utilize the space available among dead coral stands and the surrounding coral rubble substrate. They comprise a group of organisms largely overlooked in biodiversity estimates because they are hard to collect and identify, and their collection causes disturbance that is unsustainable in light of widespread reef degradation. Artificial substrate units (ASUs) provide a better sampling alternative and have the potential to enhance biodiversity estimates. The present study examines the effectiveness of ASUs made with defaunated coral rubble to estimate the diversity of motile cryptic crustaceans in the back-reef zone of the Puerto Morelos Reef National Park, Mexico. Species richness, Simpson's diversity index, Shannon–Wiener index and the composition of assemblages were compared between ASUs and samples from the surrounding coral rubble substrate. A combined total of 2,740 specimens of 178 different species, belonging to five orders of Crustacea (Amphipoda, Cumacea, Isopoda, Tanaidacea and Decapoda) were collected. Species richness was higher in the surrounding coral rubble and Shannon–Wiener and Simpson indexes were higher in ASUs. Species composition differed between methods, with only 71 species being shared among sampling methods. Decapoda was more speciose in ASUs and Peracarids in the surrounding coral rubble. Combining the use of ASUs with surrounding rubble provided a better inventory of motile cryptic crustacean biodiversity, as 65% of the species were represented by one or two specimens.

## INTRODUCTION

Estimating the biodiversity of coral reefs is challenging as many invertebrate species are rare, small, and inhabit microhabitats that are difficult to access. This is especially true of cryptofauna, which are a major component of the biodiversity of coral reefs that are hard to estimate (*Reaka-Kudla, 1997*; *Small, Adey & Spoon, 1998*), with the subphylum Crustacea being one of the most abundant and speciose groups. Its representatives occupy cracks, crevices and cavities within the reef, ranging from a few millimeters to several

centimeters in diameter, including coral framework, bioerosion galleries, and the interstices between large clasts in deposits of skeletal rubble (*Hutchings & Weate, 1977*; *Peyrot-Clausade, 1980*; *Reaka-Kudla, 1997*). Skeletal rubble is common on coral reefs that are impacted by tropical cyclones and is generated when storm and hurricane waves destroy live coral stands on the shallow inner shelf, and deposit the fragmented corals as a layer of coarse rubble covering the shallow reef zones (*Blanchon, Jones & Kalbfleisch, 1997*). In Caribbean fringing reefs, coral sand and rubble produced during these events is deposited mainly over the crest and the back-reef causing a retrograde accretion through time (*Blanchon et al., 2017*).

Skeletal rubble deposits are reported to be colonized by cryptic crustaceans in as little as 2–4 weeks (*Takada, Abe & Shibuno, 2007*), as they provide microhabitats, feeding areas, and protection against predation (*Moran & Reaka-Kudla, 1991*; *Buhl-Mortensen et al., 2009*; *Humphries, La Peyre & DeCossas, 2011*). Yet cryptic crustaceans inhabiting coral rubble have been largely overlooked in biodiversity estimates because individuals are hard to collect and identify. Furthermore, their collection is commonly destructive and involves disturbance to the collection site, which is incompatible with coral reef health and prohibited in marine protected areas.

Artificial substrate units (ASUs) are fabricated structures that mimic the characteristics of natural habitats (*Walker, Schlacher & Schlacher-Hoenlinger, 2007*). Their design can provide high spatial diversity, they are easy to place, recover, and relocate, and can provide a standardized sampling effort, allowing direct comparison between different sites (*Chapman, 2002*; *Takada, Abe & Shibuno, 2007*; *Baronio & Bucher, 2008*; *Takada et al., 2016*). ASUs can also be tracked over time to study recruitment and succession processes (*Perkol-Finkel & Benayahu, 2005*), and the response of biota to environmental gradients or short-term disturbances (*Walker, Schlacher & Schlacher-Hoenlinger, 2007*).

Several types of ASUs have been developed to study the biodiversity of hard bottom marine habitats (*Plaisance et al., 2011*; *Enochs et al., 2011*; *Takada et al., 2016*). Artificial Reef Matrix Structures, for example, are ASUs made of affordable materials which are designed to mimic large head corals (*Zimmerman & Martin, 2004*). By contrast, ASUs designed to study motile cryptofauna diversity commonly employ mesh trays filled with defaunated coral rubble, which is reported to have the highest species richness, compared to live or recently dead coral (*Enochs & Manzello, 2012*). This type of ASU has been employed on Pacific reefs (*Enochs et al., 2011*; *Takada, Abe & Shibuno, 2012*; *Takada et al., 2016*), but has been used to a lesser extent in the Caribbean, despite the fact that coral rubble is an abundant substrate and plays an important role in harboring diverse cryptofaunal communities, including fish (*Choi & Ginsburg, 1983*; *Gischler & Ginsburg, 1996*; *Valles, Kramer & Hunte, 2006*). In order to determine their efficiency, however, data derived from their employment needs to be compared with data obtained through other sampling methods.

In this study, we evaluate the efficiency of ASUs made with plastic mesh-bags filled with defaunated coral rubble as a means of obtaining the crustacean motile cryptofauna diversity and improve the species inventory in the back-reef zone of a Mexican Caribbean

 

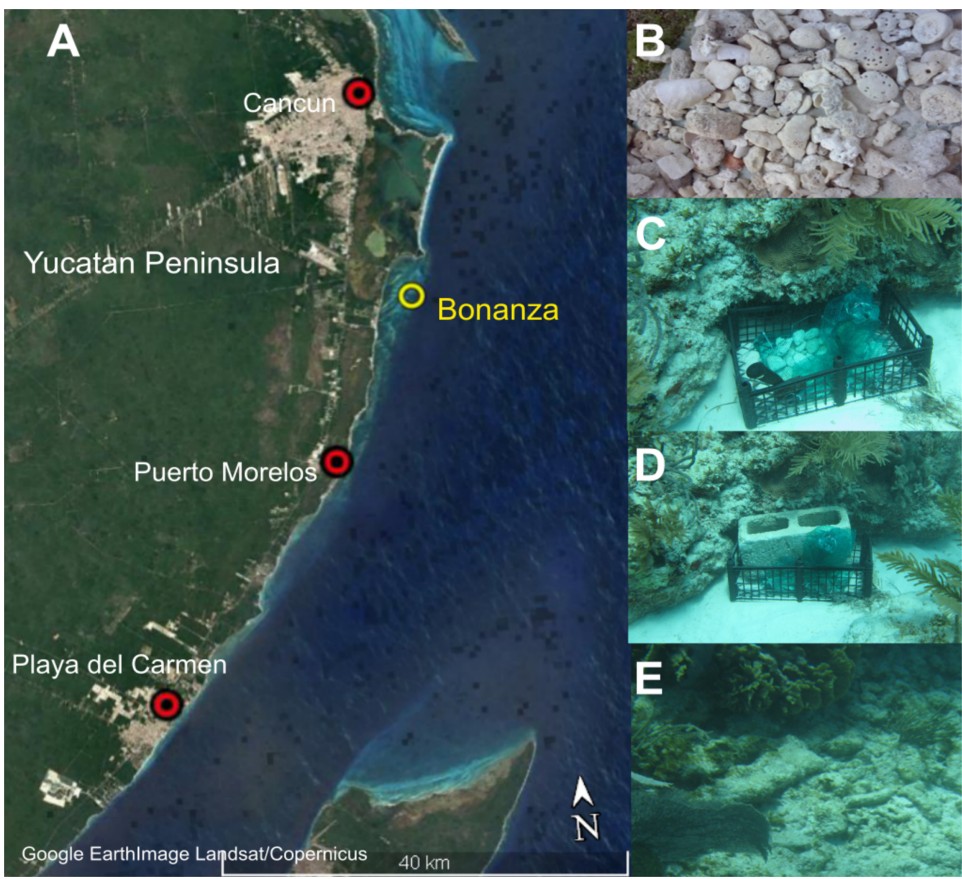

**Figure 1 Study area and method.** Study area in the northeastern Caribbean coast of the Yucatan Peninsula: (A) Google Earth image showing the location of Bonanza reef site; (B) Coral rubble collected from the beach; (C) Artificial substrate unit placed inside a vented plastic crane in the back-reef at ~3 m deep; (D) Artificial substrate unit anchored to the seafloor with a concrete block; (E) Scattered coral rubble in the back-reef zone. Map data: SIO, NOAA, U.S. Navy, NGA, GEBCO; Image; Landsat/Copernicus.

reef, where the diversity of cryptic crustaceans in coral rubble has been reported previously (*Monroy-Velázquez, Rodríguez-Martínez & Alvarez, 2017*).

# MATERIALS AND METHODS

## Study site

The study was carried out in the back-reef zone of the Bonanza reef site (20°57′58″ N, 086°48′27″ W; Fig. 1A), within the Puerto Morelos Reef National Park, in the Mexican Caribbean. The site is characterized by well-developed back-reef and crest zones, and a reef-front with limited structural relief and only small (<50 cm) scattered coral colonies (*Jordán-Dahlgren, 1979*). Between the reef and the shore, lies a reef lagoon (~2.5 km wide) colonized by seagrasses and macroalgae. The back-reef environment at Puerto Morelos is the main zone of active coral growth at present and is dominated by *Acropora palmata*, *Orbicella* spp., *Pseudodiploria* spp., *Siderastrea siderea*, *Agaricia agaricites*, and *Porites astreoides* (*Caballero-Aragón et al., 2020*), whereas the crest zone is dominated by

*A. palmata* and *Millepora complanata* (*Jordán-Dahlgren, 1979*). After tropical storms and hurricanes, a large amount of skeletal detritus from these coral species accumulates in the back-reef. Based on historical evidence, 27 hurricanes have passed within 50 km of the town of Puerto Morelos between 1852 and 2019, with Hurricanes Gilbert (1988) and Wilma (2005) being the most intense (*National Hurricane Center, 2020*). The site is also under the influence of trade winds, which are interrupted by mild cold fronts for periods of 3–10 days in the winter (*Ruiz-Rentería, Van Tussenbroek & Jordán-Dahlgren, 1998*). The Yucatan current flows northward along the narrow shelf and, during the trade wind season, its superficial waters are transported into the reef area. Monthly average sea surface temperature ranges from 25.1 to 29.9 °C (*Rodríguez-Martínez et al., 2010*).

## Artificial substrate unit design

The artificial substrate unit (ASU) was designed using a plastic tray (50 cm high by 40 cm wide) within which was placed a mesh bag (with a 35 mm mesh) filled with 3 kg of coral rubble (collected from the beach behind to the study site and dried for 5 days to ensure that it was uncolonized; Fig. 1). The coral rubble selected was naturally porous and ranged in diameter from 5 to 20 cm (Fig. 1B). The crate was anchored with a concrete block to prevent its displacement by waves and currents (Fig. 1C); the block holes were open to the surface, allowing the recruitment of cryptofauna (Fig. 1D). Using scuba, two ASUs were placed on the seafloor of the back-reef zone at a depth of 3 m, in the area where coral rubble accumulates after storms and hurricanes. These ASUs were replaced every 2–3 months with fresh rubble (May, August, and November of 2013, and January of 2014); this period was selected based on the studies of *Takada, Abe & Shibuno (2007)* who showed that a period of 2–4 weeks is sufficient for the establishment of cryptofauna on coral reefs. For retrieval, each ASU was placed into a plastic bag to prevent specimen loss. At the same time, 3 kg of the same-sized coral rubble was collected in-situ from the area surrounding the ASU with an area no larger than 9 m$^2$ (Fig. 1E). Once in the boat, both bagged samples were placed in buckets containing seawater and immediately transported to the laboratory. In total eight samples were obtained from ASUs and eight from coral rubble collected in situ throughout the study. All surveys were conducted under permit DGOPA.00008.080113.0006 granted by SAGARPA (Agriculture, Natural Resources and Fisheries Secretariat) to F. Alvarez.

## Laboratory work

In the laboratory, the coral rubble obtained from the ASUs and in situ was placed in separate buckets filled with fresh water to provoke osmotic shock and force organisms out of their cavities. The residue material was sieved through a 0.5 mm mesh. Organisms were preserved in 70% ethanol and later identified to the lowest possible taxonomic level and counted. Identifications followed *Suárez-Morales et al. (2004)* for Tanaidacea, *Heard, Roccatagliata & Petrescu (2007)* for Cumacea, *Kensley & Schotte (1989)* for Isopoda, *Thomas (1993)* and *LeCroy (2000, 2002, 2004, 2007)* for Amphipoda, and *Williams (1984)* for Decapoda.

## Data analysis

Species diversity obtained using the two sampling methods was assessed using Hill Numbers of the effective number of species (*Hill, 1973*; *Chao et al., 2014*), namely species richness ($q = 0$), the exponential of Shannon entropy index, or Shannon diversity ($q = 1$), and the inverse of the Simpson concentration index, or Simpson diversity ($q = 2$). Hill Numbers and curves, and measures of sample coverage, were obtained by means of the package iNEXT in the R environment (*Hsieh, Ma & Chao, 2016*). Sample coverage is a measure of sample completeness that gives the proportion of the total number of individuals in a community that belong to the species represented in the sample (*Hsieh, Ma & Chao, 2016*). Subtracting the sample coverage from unity gives the probability that the next individual collected belongs to a species not previously collected in the sample (*Hsieh, Ma & Chao, 2016*).

To compare species composition between methods, non-metric multidimensional scaling (NMDS) ordination was employed, using the metaMDS function (Vegan package), with Bray–Curtis dissimilarity measure and 999 permutations. Assemblage compositions were computed based on presence/absence of species. Differences in composition among methods were tested by a permutational multivariate analysis of variance with 9,999 permutations, using the nonparametric ADONIS function of the Vegan package in the R environment (*Oksanen et al., 2013*).

The Importance Value Index (IVI) (*Curtis & McIntosh, 1951*) was used as a proxy to estimate the relative importance of each taxon within each substrate. The IVI of each taxon is calculated as IVI = (RA+RF)/2, where RA is relative abundance, calculated from the number of individuals per taxon with respect to the number of individuals of all species found in the assemblage, and where RF is relative frequency, estimated as the proportion of surveys where a taxon is present, normalized to the frequency of all species in the assemblage. All analyses were done in R 3.6.3 (*R Core Team, 2019*).

## RESULTS

A total of 2,740 specimens belonging to at least 178 species, encompassing five orders of Crustacea (Amphipoda, Cumacea, Isopoda, Tanaidacea and Decapoda) and 58 families were identified and recorded throughout the study. Of these, 129 taxa were identified to species, 39 to genus and ten to higher taxonomic levels. The taxonomic composition of the samples taken using the two methods is summarized in Table S1. Fifty-five species (31%) were represented by a single specimen and 60 (34%) by two specimens each. Forty percent of the species were shared among methods. Decapoda was the most speciose order, with 57 species, followed by Isopoda ($N = 48$), Amphipoda ($N = 39$), Tanaidacea ($N = 18$) and Cumacea ($N = 16$). Three species of Decapoda represent new records for the Mexican Caribbean (*Paguristes hernancortezi*, *Processa profunda*, and *Processa riveroi*). Other specimens that were rarely observed in samples, and were not included in the data analyses, were crustaceans belonging to the class Ostracoda and to the subclass Copepoda, as well as animals belonging to Mollusca, Polychaeta, and Echinodermata.

In total, 868 specimens of crustaceans, consisting of at least 116 species, were obtained from the ASUs, and 1,872 specimens, consisting of at least 133 species, were obtained from

**Table 1 Number of families, species, and individuals of five Crustacea orders.**

| Order | ASUs | | | Coral rubble | | | Total | | | Species shared |
|---|---|---|---|---|---|---|---|---|---|---|
| | F | S | N | F | S | N | F | S | N | |
| Amphipoda | 17 | 25 | 159 | 15 | 29 | 195 | 19 | 39 | 354 | 15 |
| Cumacea | 3 | 4 | 11 | 3 | 14 | 21 | 3 | 16 | 32 | 2 |
| Isopoda | 10 | 29 | 267 | 11 | 39 | 438 | 12 | 48 | 705 | 20 |
| Tanaidacea | 7 | 11 | 244 | 9 | 17 | 1140 | 9 | 18 | 1,384 | 10 |
| Decapoda | 13 | 47 | 187 | 9 | 34 | 78 | 15 | 57 | 265 | 24 |
| Total | 50 | 116 | 868 | 47 | 133 | 1,872 | 58 | 178 | 2,740 | 71 |

Note:
Number of families (F), species (S) and individuals (N) of five orders of Crustacea retrieved from artificial sampling units (ASUs) and from coral rubble collected in situ in the Bonanza reef unit of the Puerto Morelos Reef National Park in 2013–2014.

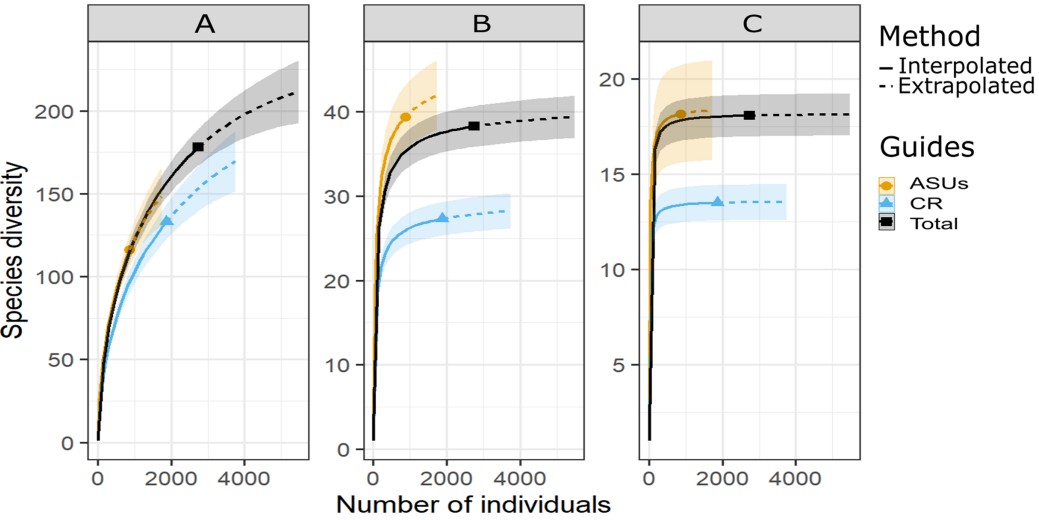

**Figure 2 Diversity of cryptic crustacea by method.** Sample-size-based rarefaction (solid lines) and extrapolation (dashed lines up to double the reference sample size) curves of motile cryptic crustacean diversity in artificial sampling units (ASU), in coral rubble collected in situ (CR), and in both methods combined (Total) based on three Hill's numbers: (A) $q = 0$ species richness, (B) $q = 1$ the exponential of Shannon's entropy index, and (C) $q = 2$ the inverse of Simpson's concentration index. The 95% confidence intervals (shaded areas) were obtained by bootstraping (300 replications). Reference samples are denoted by solid markers.               

coral rubble collected in situ (Table 1). Species richness was not significantly different between methods (confidence intervals overlap; Fig. 2) but the identity of the species differed, showing that both contribute to unique taxa; 45 were exclusive to ASUs and 62 were unique to coral rubble collected in situ. In both methods, over half of the species were represented by one or two specimens (ASUs = 57%; Coral rubble = 52%). Overall, Decapoda was more speciose in ASUs, while Isopoda, Amphipoda, Tanaidacea and Cumacea were more speciose in coral rubble collected in situ (Table 1). Regarding the number of individuals, Isopoda was the most abundant order in ASUs and Tanaidacea in

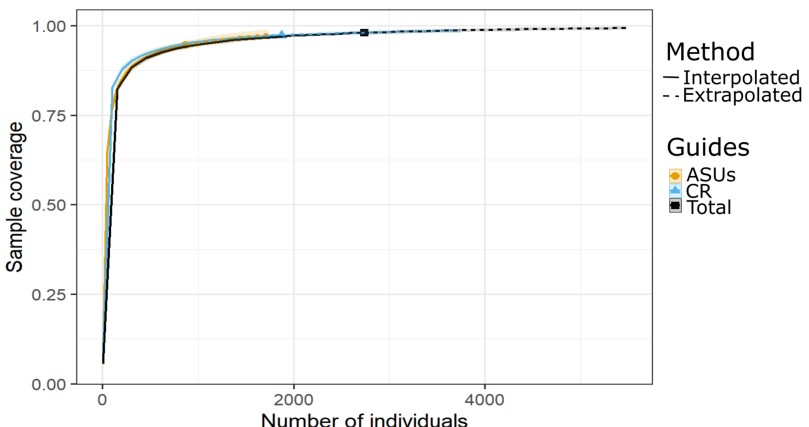

**Figure 3 Sample coverage by method.** Coverage-based rarefaction (solid lines) and extrapolation curves (dashed lines up to double the reference sample size) based on species richness of the motile cryptic crustaceans in Bonanza reef site in 2013–2014. The 95% confidence intervals (shaded regions) were obtained by bootstrapping (300 replications). Reference samples are denoted by solid markers.

coral rubble collected in situ; species of the order Cumacea were rare in samples obtained by both methods (Table 1).

Shannon and Simpson indexes were significantly higher in ASUs ($H'$ = 39.4; $D$ = 18.2) than in coral rubble collected in situ ($H'$ = 27.3; $D$ = 13.5) (confidence intervals don't overlap; Fig. 2). Rarefaction curves of species richness constructed to estimate the reliability of diversity estimates for both methods (Fig. 3) failed to reach a plateau, indicating that sample size was insufficient to reliably estimate the total number of species and thus diversity measurements for each method are conservative (Fig. 2). Estimates of sample coverage, a measure of sampling completeness, were 0.95 for ASUs, 0.97 for coral rubble collected in situ, and 0.98 when both methods were combined (Fig 3).

The nMDS plot, based on presence/absence data in Fig. 4, showed no distinct separation of the cryptic crustacean assemblages in the two methods, as confirmed by the high stress value (0.1704). Assemblages obtained from coral rubble collected from in situ samples at different periods were more similar than samples of rubble in ASUs, nevertheless the samples from both methods overlap for some sampling periods; ASUs samples from the first and last surveys were more similar to coral rubble samples collected in situ than to ASUs samples collected in the second and third surveys (Fig. 4). ADONIS test indicated that the method had a small effect, although it was significant ($R^2$ = 0.1142, $p$ = 0.0027).

According to the Importance Value Index (IVI), the dominant species differed between methods. In ASUs, the dominant species were the tanaidacean *Chondrochelia dubia* (IVI = 9.5%) and the isopod *Cirolana parva* (IVI = 6.2%), with other relatively important species being the amphipod *Elasmopus rapax* (IVI = 3.5%) and the tanaidacean *Apseudes* sp. A (IVI = 3.4%) (Fig. 5). In coral rubble collected in situ, the dominant species were the tanaidaceans *Apseudes* sp. A (IVI = 8.6%), *Paratanais* sp. A (IVI = 7.9%), *Pseudoleptochelia* sp. A (IVI = 6.2%) and *Chondrochelia dubia* (IVI = 5.2%) (Fig. 5).

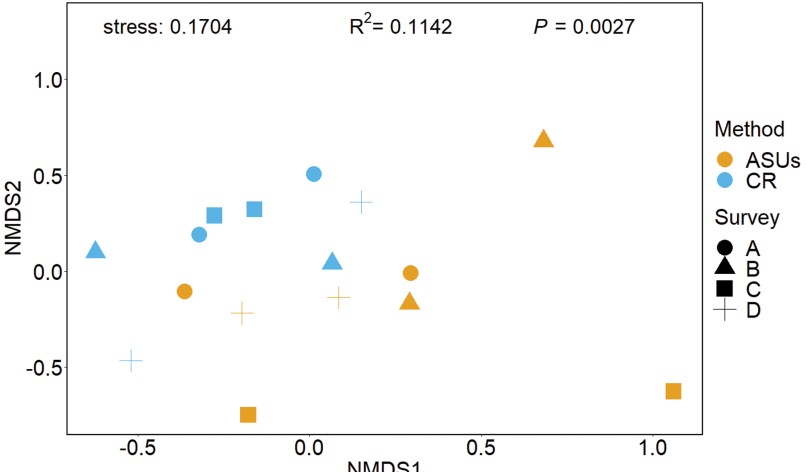

**Figure 4 Non-metrical multidimensional scaling (nMDS) ordination plot.** Non-metrical multi-dimensional scaling (nMDS) ordination plot based on Bray–Curtis similarities of motile cryptofauna communities between artificial sampling units (ASUs) and coral rubble collected in situ (CR). Letters A–D correspond to sampling periods: (A) May 2013, (B) August 2013, (C) November of 2013, and (D) January 2014.

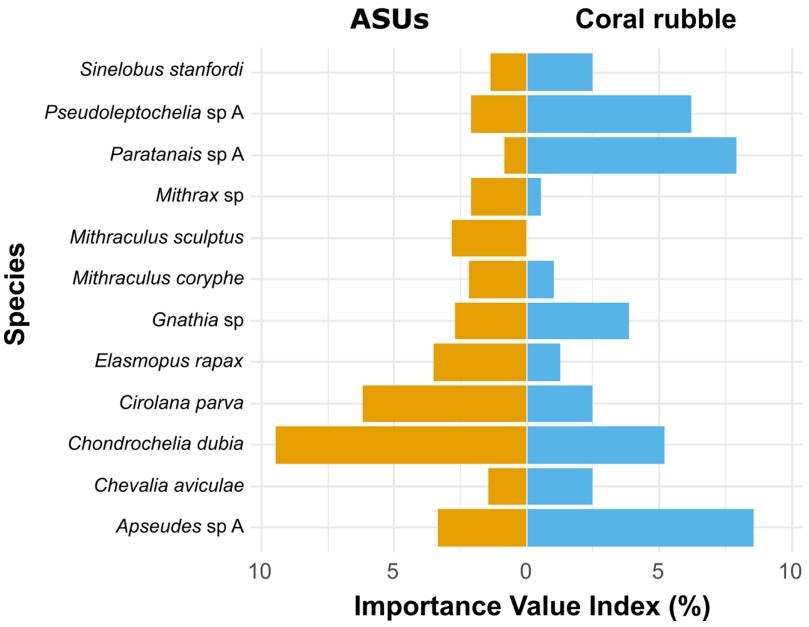

**Figure 5 Ecological Importance Value Index.** Ecological Importance Value Index of cryptic Crustacea species in artificial sampling units (ASUs) and in coral rubble collected in situ from Bonanza reef site in 2013–2014. The index is based on the relative abundance and frequency of each species with respect to the assemblage on each substrate. Only species with relative importance above 2% on either of the substrates are shown.

## DISCUSSION

Artificial sampling units (ASUs) made with fresh coral rubble and deployed in different seasons for short periods of time (2–3 months) are an effective method for improving

species inventory of motile cryptic crustaceans on Caribbean coral reefs. Using this method we recorded 116 species of this group during the 1-year study duration at the Bonanza site, 45 of which were not recorded in the surrounding coral rubble. Nevertheless, the ASUs failed to record 62 species that were unique to the surrounding coral rubble. However, the rarefaction curves of species richness for both methods failed to reach a plateau, indicating that more samples were needed to have a complete inventory.

By combining both ASUs and surrounding rubble samples, we recorded a total of 178 species, with 65% being represented by one or two individuals, and reached a sample coverage of 98% in our sampling size. The nMDS analyses showed no distinct separation of the cryptic crustacean assemblages obtained by the two methods, as samples obtained from ASUs in the first and last surveys overlapped with those obtained from coral rubble collected in situ. However, ASUs samples from intermediate surveys were dissimilar to all others suggesting that it would take more ASUs to provide estimates of the community structure recorded in coral rubble samples, or that ASUs need to be left in place for longer periods. Further studies are needed to determine at which point in time, or after how many samples, the two methods would yield similar results.

Our results support the finding of other studies which suggest that sampling of coral rubble using different techniques would render a higher taxonomic richness (*Costello et al., 2017*) and a greater potential for the discovery of new species (*Souza, Oliveira & Almeida, 2012*; *Paz-Ríos, Simões & Ardisson, 2013*). Our ASUs were more effective in sampling decapods, with 23 out of the 57 species recorded being exclusive to this method, while the surrounding coral rubble was more effective for recording unique species of Amphipoda, Cumacea, Isopoda and Tanaidacea, even though some families of these orders were only sampled by ASUs, including the Amphilochidae and Bateidae, of the order Amphipoda, and the Munnidae, of the order Isopoda. Decapod families exclusively found in ASUs were: Hippolytidae, Paguridae, Pilumnidae, Porcellanidae, Spongicolidae and Thoridae.

Differences between sampling methods in motile cryptic crustacean species richness, diversity, and assemblage composition could also be explained by the duration of time that each substrate remained underwater, and thus differences in the composition and coverage of algal turfs. Peracarids like a layer of turf algae and fine sediment particles on which to feed, while decapods were more likely recruiting to ASUs for shelter from predators or could be actively foraging within the ASUs. Coral rubble within ASUs had low algal turf coverage, as it stayed in the water for only a few months (≤3). Although biofilms formed by bacteria and microalgae can be formed within hours (*Cuba & Blake, 1983*), the composition and coverage of the algal assemblage can change significantly within months (*Fricke et al., 2011*), as opportunistic filamentous species are replaced by more competitive fleshy ones and Cyanobacteria (*Wanders, 1977*; *Fricke et al., 2011*). Given that the pattern of succession of algae can shape their communities (*Connell & Slatyer, 1977*), the absence, or low coverage, of certain algal species could have inhibited the colonization or permanence of some of the cryptic crustacean species in the ASUs. Biofilms, for example, are known to release peptides that induce the settlement of several species of sessile invertebrate larvae (*Johnson et al., 1997*) and sessile

assemblages on coral rubble may later affect the colonization of cryptic motile fauna (*Klumpp, McKinnon & Mundy, 1988*; *Kramer, Bellwood & Bellwood, 2012*). In the coral rubble collected around the ASUs, the algal-turf cover was higher, increasing habitat heterogeneity and allowing detritus to be trapped (*Danovaro & Fraschetti, 2002*). This possibly favored a higher species richness of peracarids, in particular of tanaidaceans, which are typically found in early successional stages (*Larsen & Shimomura, 2008*).

Despite its apparent permanence in back-reef environments, coral rubble cannot be seen as a static substrate, particularly in shallow reef sites, where it can be periodically reworked by currents and large wave events during storms, hurricanes, and north winds, or disturbed by fish feeding and bioerosion, among other factors, thus becoming periodically available for colonization (*Takada, Abe & Shibuno, 2007*). All these factors may drive the distribution and structure of cryptic assemblages (*Choi & Ginsburg, 1983*; *Meesters et al., 1991*) and contribute to the maintenance of high species diversity, by avoiding competitive exclusion and facilitating the colonization of less competitive species (*Enochs et al., 2011*). A higher peracarid species richness would likely be obtained by increasing the number of ASUs per survey and allowing the artificial substrate to become covered by an algal matrix before deployment.

The dominant cryptic crustacean species, as obtained by the IVI, differed between sampling methods. Coral rubble substrates were dominated by tanaids (*Apseudes* sp., *Paratanais* sp., *Pseudoleptochelia* sp. and *Chondrochelia dubia*), while in ASUs, *C. dubia* was co-dominant with isopod *Cirolana parva*, and the amphipod *Elasmopus rapax*: these species were probably opportunistic colonizers of new habitat space. Juveniles and ovigerous females of *E. rapax*, and decapods (families Alpheidae and Mithracidae) were observed in ASUs throughout the study. The dominant species in ASUs were previously reported as abundant in coral rubble substrates on the Puerto Morelos reef (*Monroy-Velázquez, Rodríguez-Martínez & Alvarez, 2017*; *Winfield et al., 2017*), suggesting that, despite their artificial nature, ASUs were not only colonized by some of the most abundant reef species, but also by rare ones too. More studies are needed to determine if ASUs have an effect on the abundance, life stage, and sizes of the individuals recruited.

Our findings show that when assessing the effectiveness of ASUs on coral reefs, or other ecosystems, care should be taken in comparing the experimental results with controls collected simultaneously from the same sample station. Changes in either of these variables can produce significant differences in species composition and abundance that will affect comparisons (*Moran & Reaka-Kudla, 1991*; *Takada, Abe & Shibuno, 2007*). Sessile encrusting or colony forming species are not expected to be common in ASUs, unless they remain in the water for several months or more (*Malella, 2007*; *Duckworth & Wolff, 2011*). Once a broad survey of the species composition of the local coral rubble has been undertaken, it is then possible to evaluate the effectiveness of ASUs. Our results show that the use of ASUs made with defaunated coral rubble is effective in detecting cryptic and rare motile crustaceans, and can help improve species inventories of this group on Caribbean coral reefs.

## CONCLUSION

Artificial sampling units (ASUs) made with defaunated coral rubble constitute a valuable tool to study the diversity of motile cryptic crustaceans in Caribbean coral reefs. Our results show that combining data from ASUs with that from surrounding coral rubble gives a more complete inventory of species, as both methods contribute unique species. ASUs gave a better estimate of diversity, whereas the surrounding coral rubble gave a better estimate of species richness. By combining both methods we recorded an assemblage of motile cryptic crustaceans composed of at least 178 species encompassing five orders at a single reef site in 1 year.

## ACKNOWLEDGEMENTS

We thank the Instituto de Ciencias del Mar y Limnología, UNAM, in particular Dr. Brigitta van Tussenbroek, for providing the laboratory space. Dive Master E. Rodríguez Soria and H. Palma provided logistic help during field work. The manuscript was greatly improved by the suggestions of Ian Enochs, Robert Toonen and one anonymous reviewer. Optical images of the study site in Fig. 1A are from Google Earth (Map data: SIO, NOAA, U.S. Navy, NGA, GEBCO; Image; Landsat/Copernicus).

### Funding

This paper is part of a dissertation submitted by Luz Verónica Monroy-Velázquez to the Posgrado en Ciencias Biológicas, Universidad Nacional Autónoma de México, in partial fulfillment of the requirements for the Ph.D. degree, with the support of the Consejo Nacional de Ciencia y Tecnología (CONACYT) who provided the scholarship number 84015. All surveys were funded by CONABIO (Grant LH010 Invertebrados del Parque Nacional Arrecife Puerto Morelos) and by DGAPA-UNAM (Grant IN205314 ¿Puede la criptofauna de crustáceos indicar el estado de conservación de un arrecife?). The funders had no role in study design, data collection and analysis, decision to publish, or preparation of the manuscript.

### Grant Disclosures

The following grant information was disclosed by the authors:
Posgrado en Ciencias Biológicas, Universidad Nacional Autónoma de México.
Consejo Nacional de Ciencia y Tecnología (CONACYT): 84015.
CONABIO: LH010.
DGAPA-UNAM: IN205314.

### Competing Interests

The authors declare that they have no competing interests.

## Author Contributions

- Luz Verónica Monroy-Velázquez conceived and designed the experiments, performed the experiments, prepared figures and/or tables, authored or reviewed drafts of the paper, and approved the final draft.
- Rosa E. Rodríguez-Martínez analyzed the data, prepared figures and/or tables, authored or reviewed drafts of the paper, and approved the final draft.
- Paul Blanchon analyzed the data, authored or reviewed drafts of the paper, and approved the final draft.
- Fernando Alvarez conceived and designed the experiments, authored or reviewed drafts of the paper, and approved the final draft.

## Data Availability

Underlying data, R code, and a reproducible record of all statistical analyses is available at GitHub: https://github.com/rerodriguezmtz/CruASUs.

## Supplemental Information

Supplemental information for this article can be found online at http://dx.doi.org/10.7717/peerj.10389#supplemental-information.

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
