# Peer review of "The use of artificial substrate units to improve inventories of cryptic crustacean species on Caribbean coral reefs"

_PeerJ, doi:10.7717/peerj.10389_

## Round 0.1 · original submission · Major Revisions

I now have reviews back from two experts in the field, and while both see considerable promise in your work, both also express the need for considerable additional work for that promise to be achieved. As you will see, one referee recommends rejection of the manuscript based on shortcomings of the approach and concerns about whether the authors have the power to test their stated hypotheses. The second referee shares a number of concerns about replication and statistical analyses but also acknowledges the enormous amount of work that already went into this study and sees value in the publication of the study with appropriate revisions.

I can see both sides, and would like to give you the opportunity to make major revisions to the manuscript that include both the additional analyses suggested and the appropriate caveats about potential shortcomings of the approach and future directions for improvement. In particular, both authors raise concerns about sampling effort and design that must be addressed in your revision, and I think most readers would appreciate a power analysis that shows exactly what is your ability to detect differences and account for the variability of the sampled communities. Even if the study lacks statistical power and the differences among collectors are purely observational, the second referee points out that these are extremely understudied communities (especially in the Caribbean) and the survey of species detected is a valuable scientific contribution in itself.

Both referees also point out that you need to make clear your specific goals for the study – is it to collect the most species, or minimize variance among collectors, or create a natural habitat with real-world assemblages of cryptic organisms? I would add that even if your initial goal was different – but you lack the replication and statistical power to convince referees that you have achieved your goal in this study – the study could be written up in a different manner that is far more likely to be acceptable. For example, if you lack the power to test your stated hypothesis, you could revise the focus of the paper as a preliminary evaluation of a method to sample this understudied community and future directions for refinement of the approach. In my experience, most referees are more supportive of that type of approach than stating and testing more ambitious hypotheses but for which the authors lack the data to draw conclusions.

As such, I am recommending that you undertake a major revision to address the comments of these two highly qualified referees who have provided you with considerable constructive feedback and recommendations for how to improve your manuscript. I look forward to seeing your revision and a detailed response to referee feedback.

Reviewer 1 ·

Basic reporting

This is a nicely written paper with only a few minor English issues. It is well constructed, covers the appropriate literature, etc.

Experimental design

I have some problems with the design of the study.
The research question is an interesting one and worthy of investigation.
Here are some specific comments.
l. 95. Nice that all organisms were removed by sun drying, but rubble in the sea is not sun-dried, usually, so the collectors won’t start to replicate the rubble zone until there is colonization by microbes and micro-algae, etc. Perhaps having left them in place for two months would have allowed that to happen, but I think that could explain some of the results.
l. 96. I have no problem with the collectors per se, although finding a way to standardize the collector might have been better. But I do wonder about placing the cinder block on the collector in the plastic crate. That will influence access to the rubble in some way, I would expect. Also, the collectors were placed adjacent to the back reef zone, but not in the back reef zone, or for that matter, on the rubble, a part of which was collected for comparative purposes (Fig. 1D).
The biggest problem with this study is that there were too few collectors to accommodate the variability of colonizers. Only two replicates each of two times. Would have been much better to deploy minimum of three collectors, but probably more, up to six, each of two times, maybe three times. I recognize that the person power to handle all those crustaceans might not be present or affordable, but I think one could also narrow the scope of the groups dealt with, or look for the 10-20 most commonly occurring species. It is nice, but not necessary to identify everything that is in the collector as many species are rare, as the authors have seen.
The second problem with the collectors is that they were placed near the back reef zone and not on or near the rubble they were intended to replicate. That most likely cut off a lot of potential colonizers. None of the peracarids have water borne larvae, and some, like the cumaceans, tanaids, etc., tend to be sedentary unless conditions get bad enough to make them move.
The data in Table 1 also suggests that the different crustacean groups respond to the relative newness of the rubble in the collectors in different ways. Cumaceans and taniads like a nice layer of microalgae and fine sediment particles on which to feed. On the other hand, the decapods are more likely looking for space in which to live away from predators.
l. 155-158. There should be some discussion of the taxa that were more abundant in the collectors vs not. Why are they there? I note that tanaids and gnathiids and a few other crustaceans were more abundant in the collectors.
l. 156. No confidence intervals visible in the Fig. 2
l. 193 ff. That is probably due to the small amount of rubble material, thus limiting the habitat space. More rubble collectors would undoubtedly collected more of the species known from the area.
l. 198-199. I don’t agree with this sentence at all. Peracarids are not typically found in early successional stages of habitats. Some species are, but the vast majority have very specific habitat requirements. In studies of recovery of bottom habitats after trawling, for example, it has been shown that cumaceans are the last group to arrive, long after the worms. Same with a number of the amphipod taxa.
l. 202-205. Yes, but it is also likely that rubble gets well “glued” to the bottom as it were during periods of relative calm due to microbial and microalgal growth. Those periods can last for months or, in the absence of strong winter storms or hurricanes, maybe even years. During those periods one would expect the rubble community to reach some state of stability with a different cryptofauna than what might be expected show up in rubble baked in the sun and placed in bags.
l. 212-226. Exactly the point.
l. 230. Chondrochelia dubia is probably a weedy species. Early and fast colonizer of new habitat space.

Validity of the findings

In the end, I am not sure that these collectors do what I think the authors want them to do. That is, provide an inventory of species from the rubble without destructive sampling. Perhaps a better designed, and aged collector would do a better job. For example, the collector could be dipped in fresh water to remove all the macrofauna, then placed in bags with 0.5 mm mesh, or smaller, and then allowed to incubate in the water for a month or two to allow the turf to develop. Then they could be placed on the bottom to see what they collect. If the objective is to get a good inventory of the cryptofauna without destructive sampling, then the cryptic habitat provided needs to be a good approximation of the real thing.

Additional comments

This is a very promising preliminary study in my view. I think you have the potential to develop a cryptofauna method, similar to the ARMS for larger organisms (that also have problems in my opinion) that can be made by everyone and used all over the world. The advantage is standardization that allows comparison.
I would urge you to continue to develop your method.

·

Basic reporting

While the majority of this manuscript is well written, there are several sentences that could use reworking. Similarly, the manuscript is well-cited and the authors possess an good understanding of the literature, though there are some gaps that should be corrected, particularly with reference to prior studies that use artificial reef substrates to assess decapod diversity.

Experimental design

I am extremely impressed with the amount of work that has gone into identification of the decapod specimens collected in this study. I have some concerns with replication and the clarity of the goals, specifically the definition of what a "good" collector is. It's not clear to me if the aim is to collect the most species or minimize variance, or create a natural habitat with real-world assemblages of cryptic organisms.

Validity of the findings

Please see specific comments below with respect to replication and recommended additional analysis, as well as clarifications.

Additional comments

This paper examines communities of cryptic decapod crustaceans associated with rubble and artificial collectors in order to assess the biodiversity of this community, as well as the efficacy of the aforementioned methodologies. This manuscript focuses on an extremely understudied community, especially in the Caribbean where they have been sampled less than in the Pacific. As such, this paper represents an important contribution and great effort was taken to identify and process so many individuals from so many species. I have several concerns pertaining to sampling effort and design, as well as the results and corresponding conclusions. I hope and trust that they can be sufficiently addressed before publication.

Line 1. Suggest: “Are artificial substrates effective tools for estimating crustacean cryptofauna diversity on coral reefs?”
Line 16. They comprise a group of organisms largely overlooked in biodiversity estimates because they are hard to collect and identify.
Line 18. That may not be sustainable in light of the widespread degradation of coral reef habitat.
Line 20: examines
Line 23: the Simpson’s … and the Shannon-Wiener
Line 24: Separate species or taxa? If not all identified to species, could say at least 178 different species, belonging to five orders of Crustacea were collected.
Line 26. Again, “taxa richness” should be substituted with “richness” or “species richness”
Line 28: reword
Line 31: “and sampling of animals associated with coral rubble”
Line 38 “Cryptofauna are”
Line 38: The part of this sentence on fishes and reef-building organisms is not needed an arbitrary. There are numerous other taxa on reefs and therefore highlighting just these two groups is ineffective.
Line 40: largest is not specific, you mean most biodiverse? Most speciose? Most abundant?
Line 41: replace “either in” with “including within gaps”
Line 47-50: This sentence needs to be cleaned up, perhaps broken into two
Line 53: Technically, I would say that Enochs et al. 2011 was more sustainable than Enochs and Manzello 2012 or Enochs 2012 (Marine Biology, not cited) in that it just used rubble in bags, that were then returned. The other studies involved sampling of framework and are therefore more destructive, though similar to this study, rubble was a large component of that...
Line 58: I think it is critical that you incorporate the work of Plaisance and others that have used NOAA and Smithsonian artificial reef monitoring structures (ARMS) to monitor cryptofauna biodiversity (primarily decapods like this study). These structures are probably the most widely-used, ubiquitous structures around the world, and are definitely worth discussing in this paper.
Line 62: Inappropriate use of the word factual.
Line 63-64: This makes it seem like the references are about artificial collectors that use rubble, rather than just the rubble itself.
Line 62-68: It strikes me that Enochs et al. 2011 employs artificial bags of rubble, directly as discussed here, rather than references to the rubble. There are also other studies that have used rubble bags as well, both in the Pacific and to a lesser extent in the Caribbean. I would suggest highlighting them as you are introducing the topic here.
Line 95: Are there permitting issues with this type of collection, drying, and redeployment? Numerous organisms naturally grow on this substrate and aerial exposure can be somewhat destructive (and smell bad).
Line 96: Are there shading concerns from the block?
Line 98: From this I am interpreting that the effective sample size is two for each of these timepoints and eight in total? Please clarify here and in the text. This is a little lower than I am comfortable with given the high variability observed within this community (especially over time) and may have ramifications for statistical analysis.
Line 101: Can you report the average depth of the rubble collected? Not water depth, but the actual depth of the layer of rubble on the sea floor. 50cm high rubble bag collectors may provide more shelter per unit area, but they also may have more restricted water exchange, and less epibenthic flora/fauna for organisms to feed on. In essence, if you sample three kilograms of rubble, how spread out of an area are you talking about vs. the 40 cm diameter of the collector?
Line 101: Can you speak to the structural similarity, perhaps size frequency distribution or volume of the rubble used in bags vs. collected? Were they the same general sizes? I think this is important for exploring the similarity of the approaches, as well as differences in the habitat/sheltering potential. Also, a detailed description of the rubble is necessary for general understanding of the habitat sampled. What species of coral and sizes? Was it highly eroded or recently dead and intact? Was it covered with algae or sponges, or bare and uncolonized?
Line 108: Was the rubble reused in the next deployment of the bags? If so, was there mortality of the sessile organisms or did that community mature over the subsequent sampling events throughout the year.
Line 119: Is there a citation for Hill numbers? I am not familiar with them.
Line 123: Why extreme transformation?
Line 126: This statement isn’t clear to me? All time points were pooled or they were pooled only within time points?
Line 127: P values equal to or less than
Line 137: great!
Line 141: wording
Line 143: Wow, well done.
Line 144: The taxonomic composition of the samples taken using the two methodologies
Line 152: , as well as animals belonging to Mollusca, Polychaeta, and Echinodermata
Line 155 and Line 163: Are you confident in the statistical power and sample sizes used to assess this or do you think the lack of significance is reflective of replication
Line 157. nMDS or other multivariate ordination would be an effective method to assess variation in community composition, showing overlap and separation between the two methodologies.
Lines 160: This underscores my question of habitat depth and the similarities in substrate porosity/ shelter potential
Line 185: reword
Line 189: I don’t agree with this. Just because they are not common in the samples does not mean that they were transient or opportunistic. Maybe they are rare, or their preferred food source wasn’t on the rubble. Maybe sampling effort wasn’t high enough to collect patchily distributed communities. There are lots of potential reasons this could be the case
Line 208: et al.
Line 209. Great paragraph and discussion
Line 218, not just algae. Other sessile organisms may be important food sources as well.
Line 224: Do you have data on this?
Line 228: I think just tanaids is preferable as their classification as shrimps is colloquial and factually incorrect.
Line 233: Yes abundant species are recruiting to them, but also rare. I also think the last part of the sentence is not necessarily correct. There could still be an effect in their colonization as their abundances, sizes, etc. could be different.
Line 237: reword
Line 241: How do you know that great replication or longer deployment time would have yielded the same results? Do you have enough data to state that the substrate/collector is the difference, not the amount of material or time of community development/succession?
Line 241: I think you need to be more clear on what the actual goal is. Are you trying to collect the maximum number of decapod species possible or are you trying to determine if the 2-month collector deployment reflects the same species composition, diversity, and abundances? These are different goals and the conclusions as to the effectiveness of each are different. It also strikes me that part of the “effectiveness” of the collectors is their community variance and sensitivity to environmental differences/change. Are successive collections similar enough that they can be used to detect the influence of environmental gradients? That would be very valuable or “effective” and wouldn’t necessarily result in the highest number of species collected. I think community similarity should be investigated further, and I think ordination would be helpful.
Line 266: substitute access for sample
Line 254: What about flow, porosity, and benthic cover as investigated in Enochs et al. 2011?

---

## Round 0.2 · Minor Revisions

I have now heard back from both referees who agree that the manuscript is greatly improved from the initial submission, and that it should become acceptable for publication following some minor revisions. I concur and expect that you should be able to address these remaining issues quite easily. I do not expect that this will require additional review, and look forward to seeing your revised manuscript.

Reviewer 1 ·

Basic reporting

This second version is much more modest in its goals. The paper is well written with only a few sentences that need checking for English. There were no page or line numbers but an example is "Cumacea was...".
Literature sufficient, article structure very professional, paper easy to read.
Figure legends are missing from revised draft.

Experimental design

The authors have reduced their goals due to the very limited sampling.
The design obviously cannot be changed at this time, so how to make the best of the data they have. I think they have done what can be done but need to add a few more thoughts to the Discussion.

Validity of the findings

The MDS tells us something the authors could point out. They do say that the ASU and the CG provide mostly different fauna lists. They admit that could be due to their limited sampling. But they should also talk about sampling issues for the cryptofauna in general. Why would the two sampling methods produce such differences in faunal composition. I disagree with the authors that there is much overlap of the two faunal lists, but that's not as important as understanding the differences. There is some discussion of that but that discussion avoids a question of how many samples would it take to get all the fauna.

Figure 3 suggests that it would not take many in order to account for most of the species. The rarefaction curve, however, is an estimate of species accumulation. The MDS was run on species abundance data, not presence-absence data, and it shows that the ASUs don't help to estimate community structure of the cryptofauna, i.e., there is very little overlap of the two data sets. That is a difference that I think should be pointed out. Might also be worth running the MDS on presence data. I suspect there would be a greater overlap. The raw counts for the samples are not provided in the supplemental data so that can't be checked. In any case, it does seem like it would take a lot of ASUs to provide estimates of the community structure as seen in the coral gravel samples.

Additional comments

This is much better and more modest. I think the idea of using ASUs to sample this habitat is an interesting idea, but you need to strengthen your case that it will be better than sampling the coral gravel itself, which is the conservation goal (I am very interested in conservation but I am not so sure that much is lost by sampling the coral gravel... you could take 10 samples of 250 cc and not make much of an impact on that habitat, although that might not be legal, as you point out). To replace that sampling, many more ASUs will need to be used.

All that said, I am pleased that someone is paying attention to the cryptofauna. Please keep working on this topic.

·

Basic reporting

I think this manuscript is greatly improved, minor comments are provided directly below.

Experimental design

The shift in direction and design of the current study are entirely appropriate and I commend the authors for making the difficult decision to do so, as well as the work involved with revising the paper. I think it has been a worthwhile endeavor.

Validity of the findings

See comments below. Findings are sound.

Additional comments

Suggest: The use of artificial substrate units to the improve species inventories of …
-or- improve inventories of cryptic crustacean species on …
Line 23: can cause disturbance that is unsustainable in light of widespread
Line 25: coral rubble
Line 27: the composition
Line 31: rubble not gravel, here and throughout.
Line 34: with sampling of the surrounding…
Line 44: comma after citation, before “with”
Line 44: Suggest checking out the following reference: Small AM, Adey WH, Spoon D (1998) Are current estimates of coral reef biodiversity too low? The view through the window of a microcosm. Atoll Res Bull 458:1-20.
Line 55: write out numbers less than 10… here and throughout.
Line 58: crustaceans
Line 61: Not entirely fair to single these authors out as the only destructive samplers. Numerous people have sampled rubble (including Monroy-Velazques et al. 2017) and live corals as they did.
Line 72: Wrong paper. Enochs et al. 2011 used artificial rubble bags.
Line 78: Suggest checking out: Valles H, Kramer DL, Hunte W (2006) A standard unit for monitoring recruitment of fishes to coral reef rubble. J Exp Mar Biol Ecol 336:171-183.
Line 79: Suggest different word than “dominant.” Not clear what you mean. I’m guessing most abundant substrate, though the references cited indicate “play an important role in harboring diverse cryptofaunal communities.
Line 116: Length? Width? Circumference?
Line 136: Need to give brief description here so the reader can figure out what you did without reading the other paper. Okay to provide less details and point the reader to the other paper but still need a synopsis.
Line 146. Suggest new sentence
Line 149: I would suggest sqrt transformation to decrease the potential for extremely abundant species to strongly skew your assemblage comparison. This is pretty common practice. It may also help to deal with the very high stress in your nmds plot.
Line 188: italicize.
Lines 194: and 205: Odd spacing.
Line 235: Looking for space is conversational. Suggest “recruiting to ASUs for shelter from predators” …though previous studies have found abundant cryptic predator populations. I also think that it is entirely possible that herbivorous, detritivorous, and carnivorous species were actively foraging within the ASU’s. Trophic consideration of the decapods considered may be helpful in making this case.
Line 251: reference?
Line 278: Not sure if I follow. What if the most abundant species are dependent on a food source that takes time to recruit to the ASU’s?

---

## Round 0.3 · accepted · Accept

Thank you for your revised manuscript. Both referees were already supportive of the last version and suggested only minor revisions to your submitted manuscript. Having read through this, I see that you have addressed all the minor suggestions made by each of the reviewers. Therefore, I do not see any reason to delay this process any longer and am happy to move your paper forward into production.